# The Involvement of Thalamic Nuclei in Schizophrenia Patients with and Without Auditory Verbal Hallucinations: A Brain Morphometry Study

**DOI:** 10.3390/brainsci15090914

**Published:** 2025-08-25

**Authors:** Fahad H. Alhazmi

**Affiliations:** Department of Diagnostic Radiology, College of Applied Medical Sciences, Taibah University, Medina 42353, Saudi Arabia; fhdhazmi@taibahu.edu.sa; Tel.: +966-556758688

**Keywords:** schizophrenia, auditory verbal hallucinations, thalamic nuclei, MRI, PSYRATS

## Abstract

Background: The thalamus, together with its associated nuclei and thalamocortical pathways, is crucial in understanding the neurobiological mechanisms underlying auditory verbal hallucinations (AVHs) in schizophrenia (SCZ). Purpose: The study investigates the role of thalamic nuclei in schizophrenia patients experiencing auditory verbal hallucinations (AVHs). Methods: A total of 21 healthy controls (HC), 22 schizophrenia patients without auditory verbal hallucinations (SCZ_AVH−), and 22 schizophrenia patients with auditory verbal hallucinations (SCZ_AVH+), aged between 19 and 65 years, were recruited. They underwent MRI scans, and participants in the SCZ_AVH+ group completed the assessment of the severity of different dimensions of auditory hallucinations and delusions using the psychotic symptom rating scale (PSYRATS). High-resolution T1-weighted imaging was utilized to analyze the volumes of the thalamic nuclei. Results: The volumetric analysis of the thalamic nuclei indicated a significant reduction (t = 2.64, *p* = 0.03) in the right medial geniculate nucleus (MGN) volume in the SCZ_AVH+ group (0.08 ± 0.01 cm^3^) compared to the SCZ_AVH− group (0.09 ± 0.01 cm^3^). Also, the SCZ_AVH− group (0.03 ± 0 cm^3^) showed a significant increase (t = −2.73, *p* = 0.02) in right habenular nucleus (HN) volume compared to the HC group (0.02 ± 0 cm^3^). Significant correlations were observed between the volume of the left MGN and psychotic ratings (r= −0.5), as well as between the volume of the right HN and psychotic ratings (r= 0.56). Conclusion: The volumetric changes are observed in both SCZ_AVH− and SCZ_AVH+ groups, mainly in the thalamic nuclei. Structural deficits in the MGN may distinguish schizophrenia patients with AVHs from those without.

## 1. Introduction

Schizophrenia (SCZ) is considered a severe and long-lasting mental health disorder marked by hallucinations, delusions, and disorganized thoughts and behaviors. It affects approximately 24 million people, or 1 in 300 worldwide [1], typically developing in men during late adolescence to early adulthood and in women between their early 20s and early 30s [2]. It is hypothesized that SCZ results from a complex interaction of genetic predisposition and environmental factors, rather than a single cause [3]. Symptoms of SCZ can be categorized as positive (hallucinations and delusions), negative (lack of motivation and social withdrawal), and cognitive (thinking and memory) symptoms [4].

Auditory verbal hallucinations (AVHs) are common in schizophrenia, affecting up to 80% of diagnosed individuals [5]. The primary characteristics of SCZ with AVHs include acoustic and linguistic properties, frequency, control, inner-outer localization, content, personification, appraisals, and changes over time [6]. Although AVHs are not unique to SCZ, it is assumed that they are strongly associated with the disorder [7]. Hearing voices can be distressing and disrupt social life and work responsibilities [8].

A comprehensive understanding of auditory hallucinations in SCZ requires a multidimensional approach that includes genetic, neurophysiological, and neurocognitive research in individuals with SCZ who experience AVH. Neuroimaging research is ongoing to understand the brain mechanisms underlying AVHs in SCZ patients. Some studies suggest that impaired brain processes, such as an inability to suppress self-generated sounds, may play a role, potentially paving the way for new treatment strategies [9,10,11]. There are notable structural differences in certain brain regions involved in auditory and speech perception, as well as in language and memory processing [12]. Research studies have found a decrease in gray matter volume in the superior temporal gyrus (STG) and abnormalities in the insula, anterior cingulate cortex (ACC), inferior frontal gyrus (IFG), and subcortical structures, such as the hippocampus and thalamus [13,14,15,16,17]. It is thought that these brain regions are part of a network involved in speech production and perception [18].

The cerebello-thalamo-cortical (CTC) circuit is a neural pathway that links the cerebellum, thalamus, and cerebral cortex, playing a vital role in motor control, learning, coordination, and potentially contributing to cognitive functions [19]. The thalamus functions as a relay station, receiving input from the cerebellum and transmitting it to the cerebral cortex, which helps process and regulate the information before passing it to specific cortical regions [20]. Wei et (2022) found that there are associations between structural and functional differences in the thalamus and its connections to other brain regions in SCZ patients experiencing AVH [21]. Studies have indicated that SCZ patients with AVH may exhibit altered thalamic volume, connectivity with surrounding areas, and metabolic changes [17,22,23,24]. These abnormalities in the thalamus, a crucial relay center in the brain, seem to play a role in abnormal auditory processing, leading to the perception of voices that are not present in external stimuli.

The thalamus consists of nuclei, which are clusters of densely packed neuronal cell bodies that show functional diversity. Each side of the thalamus has four regional groups of nuclei: the anterior region (anterior ventral nucleus, AVN), the lateral region (ventral lateral posterior nucleus, VLPN; ventral lateral anterior nucleus, VLAN; ventral anterior nucleus, VAN; and ventral posterior lateral nucleus, VPLN), the medial region (mediodorsal nucleus, MN; centromedian nucleus, CN; and habenular nucleus, HN) and the posterior region (pulvinar nucleus, PN; medial geniculate nucleus, MGN; and lateral geniculate nucleus, LGN). The nuclei in the anterior region receive information from the brain’s limbic system, which is essential for emotional states such as attention, alertness, and memory formation [25]. The nuclei in the lateral region are involved in spatial navigation, limbic functions, and visual processing [26]. In the medial region, the nuclei are crucial for awareness, conscious experience, arousal, sleep, vigilance, and cognitive, sensory, and sexual functions [27]. Ultimately, the posterior thalamic nuclei play a pivotal role in sensory integration, particularly for visual and auditory information, and are also involved in attention and awareness [28].

Therefore, the thalamus, along with its interconnected nuclei and thalamocortical circuits, may play a vital role in the neurobiology of AVHs in SCZ. Abnormalities in this structure and its interactions with other brain regions might lead to the misinterpretation of internal thoughts as external voices, resulting in auditory hallucinations. This study aims to analyze the brain morphometry of thalamic nuclei in SCZ patients experiencing AVHs. It will evaluate changes in thalamic nuclei volumes and lateralization indices by comparing SCZ patients with and without AVHs. It is important to note that, so far, no studies have examined the differences in lateralization indices of thalamic nuclei volumes between SCZ patients with AVHs and those without. Additionally, the study will investigate the correlation between psychotic symptom rating scale (PSYRATS) scores in SCZ patients with AVHs and thalamic nuclei volumes.

## 2. Materials and Methods

### 2.1. Subjects

The data for the current study were derived from the OpenNeuro database (https://openneuro.org/datasets/ds004302/versions/1.0.1, accessed on 20 March 2025), with dataset identifier ds004302 and version 1.0.1. All participants provided written informed consent before participation. All study procedures received prior approval from the Research Ethics Committee of FIDMAG Sisters Hospitallers (Comité de Ética de la Investigación de FIDMAG Hermanas Hospitalarias) (Project no. PI21/00416 to EP-C), and adhered to its ethical standards for human experimentation, as well as the Helsinki Declaration of 1975, updated in 2008. This dataset is recognized for its open accessibility, public availability, and absence of usage restrictions.

The study involved 71 participants divided into three groups: 25 healthy controls (HC), 23 schizophrenia patients without auditory hallucinations (SCZ_AVH−), and 23 schizophrenia patients with auditory hallucinations (SCZ_AVH+). Each group was gender-balanced, including 17 females and 45 males, aged between 19 and 65 years, with IQ scores from 71 to 116. The dataset contains structural T1-weighted magnetic resonance imaging (MRI) data, along with assessments of the severity of various types of auditory hallucinations and delusions, measured using the psychotic symptom rating scales (PSYRATS) for each participant in the SCZ_AVH+ group [29]. All questionnaire responses were collected prior to the brain imaging sessions.

The control group consisted of 25 healthy individuals, carefully selected to match the two patient groups in terms of age, sex, and estimated premorbid IQ. Participants were screened and excluded if they reported any history of treatment with psychotropic medications beyond occasional use of night sedatives. Additionally, healthy controls were disqualified if they had a family history of major psychiatric disorders in a first-degree relative. Patients were excluded based on the following criteria: (a) age under 18 or over 65, (b) history of brain trauma or neurological disorders, or (c) evidence of alcohol or substance abuse/dependence within the 12 months before participation. All participants were right-handed. Ultimately, 23 patients from the SCZ_AVH+ group and 23 from the SCZ_AVH− group were included, after accounting for reasons such as failure to complete the scanning, excessive head movement, IQ scores below 70, inability to recall task details post-scanning, not being completely hallucination-free in the SCZ_AVH− group, and matching considerations.

### 2.2. MRI Data Acquisition

High-resolution structural brain images were obtained using a 3T Philips Ingenia scanner (Philips Medical Systems, Best, The Netherlands) (Figure 1). The Fast Field Echo (FFE) sequence was employed with the following parameters: TR = 9.90 ms; TE = 4.60 ms; Flip angle = 8°; voxel size = 1 × 1 mm; slice thickness = 1 mm; number of slices = 180; FOV = 240 mm.

### 2.3. Pre-Processing Methods

The raw MRI brain images were analyzed using volBrain Online software, an automated online system for MRI brain volumetry (https://volbrain.net/, accessed on 1 May 2025) [30]. The Deep Thalamus pipeline was used to segment the thalamic nuclei (left and right) based on T1-weighted MR images. The labeling protocol employed Morel’s stereotactic atlas of the human thalamus, which is based on multiarchitectonic parcellation [31]. To achieve a more compact segmentation of the thalamus, the intermediate space label (ISN) was included.

The high-resolution T1-weighted imaging volumes underwent standard pre-processing steps, including noise removal, inhomogeneity correction, Montreal Neurological Institute (MNI) registration, Intracranial cavity (ICC) extraction, partial volume (PV) estimation, hemisphere segmentation, cerebellum segmentation, and thalamic nuclei segmentation (Figure 2). The Spatially Adaptive Non-Local Means (SANLM) filter was used to reduce the random noise inherent in the images. The N4 bias correction method was applied to correct inhomogeneities caused by the acquisition process. Affine registration was performed to align the images with the ultra-high-resolution MNI152 space (0.5 mm isotropic resolution), using Advanced Normalization Tools (ANTs) software (versions 2.x) [32]. The resulting images had a standard matrix size of 362 × 434 × 362 voxels and an isotropic resolution of 0.5 mm. After alignment in the MNI152 space, the images were cropped to focus solely on the subvolume containing the right and left thalamus. The cropped volumes were defined using preset limits based on thalamus labels, including a safety margin of 10 voxels in each dimension to account for anatomical variability. The final cropped volume measured 76 × 91 × 79 voxels.

### 2.4. Region of Interest (ROI) Selection

The study focused on four regions that include 22 thalamus nuclei, divided into right and left hemispheres. The regions are: anterior region, with the anterior ventral nucleus (AVN); lateral region, including the ventral lateral posterior nucleus (VLPN), ventral lateral anterior nucleus (VLAN), ventral anterior nucleus (VAN), and ventral posterior lateral nucleus (VPLN); medial region, comprising the mediodorsal nucleus (MN), centromedian nucleus (CN), and habenular nucleus (HN); and posterior region, consisting of the pulvinar nucleus (PN), medial geniculate nucleus (MGN), and lateral geniculate nucleus (LGN), as depicted in Figure 3.

### 2.5. Quality Control Procedure

Visual quality control of the dataset was performed to evaluate the accuracy of the registration process between individual brain images (in native space) and the MNI template (in MNI space), as well as to assess the segmentation performance of the thalamic nuclei. After visual inspection, six cases were excluded from the original dataset because of segmentation failures. As a result, the final dataset consisted of 65 cases (21 HC, 22 SCZ_AVH−, 22 SCZ_AVH+).

### 2.6. ICV Normalization Methods

The volumes of the thalamus and thalamic nuclei were normalized to the total intracranial volume (ICV) using the following equation:Vol_adj = Vol_raw ∗ (ICV_mean − ICV_raw)

In this equation, “Vo_adj” refers to the normalized volumes of the regions of interest (ROIs), “Vol_raw” is the absolute volume of the ROIs extracted from the raw data, “ICV_mean” is the average intracranial volume, and “ICV_raw” is the individual intracranial volume extracted from the raw data. All values are measured in cubic centimeters (cm^3^).

### 2.7. Hemispheric Asymmetries Measurements

An additional parameter used in the analysis is the Asymmetry Index (AI), which measures the difference between the volumes of the right and left thalamus and thalamic nuclei, normalized by their average volume. This calculation produces positive values indicating leftward asymmetry, negative values indicating rightward asymmetry, and zero indicating no directional asymmetry. The formula for the Asymmetry Index (AI) is as follows:Asymmetry Index (AI) = (Volume L − Volume R)/(0.5 ∗ (Volume L + Volume R))

In this equation, “Volume L” refers to the volumes of the ROIs in the left hemisphere, while “Volume R” represents the volumes of the ROIs in the right hemisphere.

### 2.8. Statistical Analysis

All statistical analyses were performed using DATAtab Software Inc. (2024): Online Statistics Calculator, provided by DATAtab e.U. Graz, Austria (URL: https://datatab.net (accessed on 15 May 2025)). The analysis used normalized volumes obtained from the volBrain program.

All analyses considered *p*-values below 0.05 as statistically significant, leading to the rejection of the null hypothesis. Chi-square analysis (X^2^) was employed to detect significant differences in categorical variables. ANOVA was used to compare demographic features and volumes of the thalamus and its nuclei across study groups. A post hoc test with Bonferroni correction was conducted to identify specific group differences. Additionally, Pearson correlation tests (r) evaluated the relationship between the Psychotic Symptom Rating Scale (PSYRATS) scores and volumetric measurements.

Multiple linear regression analysis was used on the brain volumetric measurements, with these measurements as dependent variables and demographic factors (age, gender, study groups, and IQ) as independent variables. Bonferroni correction for multiple comparisons was applied by dividing the *p*-values by the number of tests, resulting in a significance threshold of *p* = 0.05/4 = 0.0125. All results were reported with a significance level of *p* < 0.0125 after Bonferroni correction.

## 3. Results

### 3.1. Demographics

The demographic characteristics of the study participants are detailed in Table 1. A one-way ANOVA and a Chi-square test were conducted to evaluate differences in age, IQ, and gender among the study groups. The results showed no significant differences in participants’ age (F(2, 62) = 2.12, *p* = 0.128), gender (F(2, 62) = 2.12, *p* = 0.128), or IQ (F(2, 62) = 0.77, *p* = 0.469) across the HC, SCZ_AVH−, and SCZ_AVH+ groups.

### 3.2. Comparison of ICVs Between Study Groups

The volumetric analysis of the total ICVs showed no significant differences (F(2, 62) = 2.11, *p* = 0.13) between the study groups, as displayed in Table 2.

### 3.3. Comparison of Volumetric Measurements of Thalamus Nuclei Segmentation Between Study Groups

The volumetric analysis of the thalamic nuclei revealed notable differences in the normalized volume of the right medial geniculate nucleus (MGN) (F(2, 62) = 4.2, *p* = 0.01) and the right habenular nucleus (HN) (F(2, 62) = 3.7, *p* = 0.03) across the various study groups, as shown in Table 3. Post hoc comparisons demonstrated a significant reduction (t = 2.64, *p* = 0.03) in the right MGN volume in the SCZ_AVH+ group (0.08 ± 0.01 cm^3^) compared to the SCZ_AVH− group (0.09 ± 0.01 cm^3^), as shown in Table 4 and Figure 4. However, there were no significant differences between the HC and SCZ_AVH+ groups, nor between the HC and SCZ_AVH− groups. Furthermore, the SCZ_AVH− group (0.03 ± 0 cm^3^) showed a significant increase (t = −2.73, *p* = 0.02) in right HN volume compared to the HC group (0.02 ± 0 cm^3^), as shown in Table 4 and Figure 5. No significant differences were found between the HC (0.02 ± 0 cm^3^) and SCZ_AVH+ groups, or between the SCZ_AVH− (0.03 ± 0 cm^3^).

### 3.4. Comparison of Hemispheric Laterality of Thalamus and Thalamic Nuclei Volumes Between Study Groups

The analysis of hemispheric laterality showed that no significant differences were found in the Asymmetry Index (AI) across the thalamus and all thalamic nuclei among the groups, as shown in Table 5.

### 3.5. Correlation Between the Severity of Delusions and Hallucinations and the Volume of Thalamic Nuclei in Schizophrenia Patients with Auditory Hallucinations

Regarding the relationship between the severity of delusions and hallucinations and the volume of thalamic nuclei in the SCZ_AVH+ group, significant correlations were observed between the volume of the left MGN and the psychotic ratings (r = −0.5), as well as between the volume of the right HN and psychotic ratings (r = 0.52), as shown in Figure 6 and Figure 7.

### 3.6. Regression Model

Multiple regression analysis indicated that the relationship between total volumetric measurements and demographic characteristics is significant in the thalamus (F(5, 62) = 4.64, *p* = 0.001) and specific thalamic nuclei, namely PN (F(5, 62) = 3.97, *p* = 0.003) and HN (F(5, 62) = 2.94, *p* = 0.01). Age was significantly associated with volumetric changes in the thalamus (β = −0.31, *p* = 0.01), VPLN (β = −0.3, *p* = 0.01), and PN (β = −0.36, *p* = 0.003). Additionally, IQ was related to volumetric changes in the CN (β = −0.34, *p* = 0.009) and MN (β = 0.3, *p* = 0.01), as shown in Table 6. However, there is no significant interaction (*p* > 0.0125) between demographic characteristics and study groups regarding changes in volumetric measurements. This suggests that the effects of demographic characteristics and study groups on volumetric measurements can be considered independently.

## 4. Discussion

This study aims to evaluate the volumes of thalamic nuclei in individuals with schizophrenia, differentiating between those who experience auditory verbal hallucinations (AVH) and those who do not. The main findings include: (1) The SCZ_AVH+ group exhibited a notable decrease in the volume of the left medial geniculate nucleus (MGN) compared to the SCZ_AVH− group; (2) the SCZ_AVH− group showed a significant increase in right HN volume compared to the HC group; (3) a significant negative correlation was found between left MGN volume and psychotic ratings in the SCZ_AVH+ group; (4) a significant positive correlation was observed between right habenula nucleus (HN) volume and psychotic ratings in the SCZ_AVH+ group; and (5) no significant interaction was observed between study characteristics and groups in relation to changes in volumetric measurements.

Previous research studies have reported on the relationship between thalamic volume changes and schizophrenia, both with and without AVH. A comparable study by Pérez-Rando et al. (2022) identified several volumetric changes in thalamic nuclei in schizophrenia patients with AVH and included PSYRATS scores; however, they did not analyze lateralization [33]. Also, Ysbæk-Nielsen et al. (2024) examined differences in grey matter volumes among the cortex, anterior cingulate (ACC), superior temporal gyrus (STG), hippocampi, and thalamus between schizophrenia patients and healthy controls, revealing decreased grey matter volumes in schizophrenia patients [34]. Additionally, a postmortem study indicated a significant volume reduction in the thalamus in schizophrenia patients compared to controls [35]. A voxel-based morphometry study also found that reduced grey matter volumes in the left putamen and thalamus correlated with delusions in schizophrenia [23]. Furthermore, a study combining dynamic causal modeling and voxel-based morphometry revealed decreased blood-oxygen-level-dependent (BOLD) signals in the fronto-thalamic network during the Stroop task, along with reduced connectivity between the thalamus and the ACC and diminished white matter volume in the thalamus and frontal cortex [36]. These findings are consistent with the results of the current study, which also show a link between the thalamus and schizophrenia.

The MGN plays a critical role in auditory processing, serving as a key relay point in the auditory pathway by transmitting information from the inferior colliculus in the midbrain to the auditory cortex. Genetic factors, such as the 22q11.2 microdeletion syndrome—a known risk factor for schizophrenia—have been linked to disruptions in synaptic transmission between the MGN and auditory cortex [37]. Studies have shown that the MGN exhibits structural and functional abnormalities in schizophrenia patients with AVH, including reduced MGN volumes [33] and altered connectivity between the MGN and auditory cortex [37], potentially contributing to auditory hallucinations. These findings are consistent with the current study, which revealed reduced left MGN volume in schizophrenia patients with AVH compared to those without, along with a negative correlation between MGN volume and psychotic ratings. The abnormalities observed in the MGN may be associated with auditory hallucinations, a common symptom of schizophrenia, providing insight into how internal thoughts or sensory input can be misinterpreted as external voices.

Brain hemisphere lateralization is a significant characteristic of the human brain linked to schizophrenia [38]. The current study identified no significant differences in thalamus and related nuclei volume lateralization between study groups. On the other hand, Liu et al. (2022) reported alterations in resting-state MRI asymmetry in first-episode schizophrenia patients [39]. Additionally, He et al. (2021) found abnormal lateralization in neural pathways and connectivity patterns associated with the basal ganglia and thalamus in individuals with schizophrenia [40]. This discrepancy in results could be due to methodological issues and differences in context. These differences in thalamic asymmetry may relate to the observed aberrant lateralization in neural pathways and connectivity patterns in schizophrenia, enhancing our understanding of thalamic lateralization in this context.

The HN is a small structure in the brain situated adjacent to the dorsomedial thalamus, consisting of two primary subnuclei: the lateral and dorsal habenula nuclei [41]. This nucleus serves as a link between the basal ganglia and the limbic system, which is crucial for processing reward and aversion [42]. Research studies have shown that the HN, especially the lateral HN, displays both structural and functional irregularities in individuals with schizophrenia [43,44]. Consistent with these observations, the present study found that the SCZ_AVH− group showed a significant increase in right HN volume compared to the HC group, and there was a notable positive correlation between the volume of the right habenula nucleus and psychotic ratings in the SCZ_AVH+ group. These alterations in the habenula may influence symptoms of schizophrenia by affecting reward processing, motivation, and the regulation of dopamine pathways.

The relationship between thalamic volume in schizophrenia and some clinical variables, such as antipsychotic medication, duration of illness, and age of onset, is not fully understood, and the specific effects may vary depending on the medication type, dosage, treatment duration, and individual patient characteristics. For example, Ho et al. (2011) demonstrated that antipsychotic medications can cause a slight but detectable reduction in brain tissue over time, emphasizing the need for thorough evaluation of dosage, length of treatment, and off-label applications [45]. Furthermore, a postmortem study found no significant correlations between thalamic volumes and either the duration of illness or the patients’ age [35].

This study acknowledges several limitations. First, the sample size is relatively small, which suggests that future research should be conducted with a larger group. Second, the study used only one clinical measurement (PSYRATS), and that including additional clinical variables such as age at onset, illness duration, medication status, Positive and Negative Syndrome Scale (PANSS), and Hamilton Rating Scale for Depression (HAMD) would improve understanding of the relationship between thalamic nuclei volumes and clinical variables in patient groups. Third, due to the nature of the data and study design, the power analysis for the chosen sample size was not conducted in the current study, which limits causal inference and should be addressed in future research. Finally, combining volumetric analysis with other methods, such as structural and functional connectivity assessments, would help in understanding the pathophysiology of schizophrenia with and without AVH.

## 5. Conclusions

Structural deficits in the thalamus, especially within the thalamic nuclei, are seen in both the SCZ_AVH− and SCZ_AVH+ groups. This may help distinguish schizophrenia patients with AVHs from those without. Exploring the role of the thalamus, particularly the MGN, could shed light on the mechanisms underlying schizophrenia with AVH. Further research is necessary to better understand the involvement of the MGN by combining structural and functional neuroimaging methods in larger samples.

## Figures and Tables

**Figure 1 brainsci-15-00914-f001:**
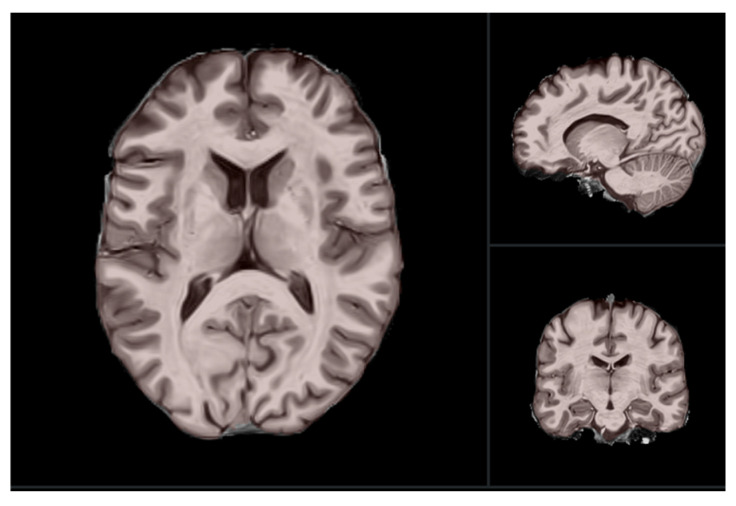
An unlabeled MRI image example for reference [subject 05].

**Figure 2 brainsci-15-00914-f002:**
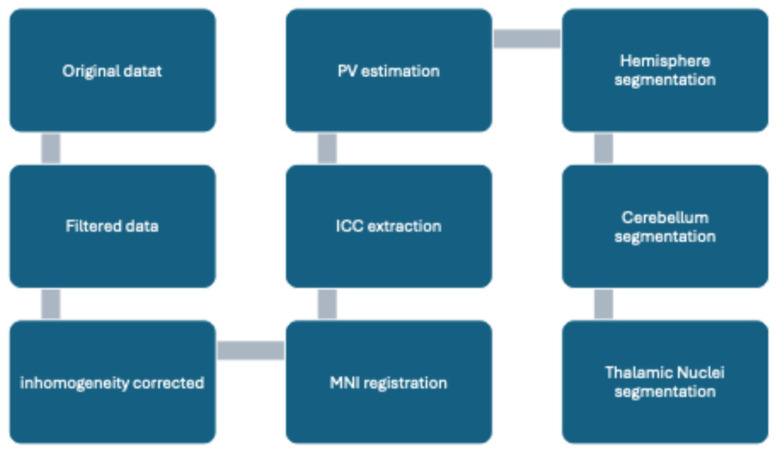
Pre-processing diagram steps of the VolBrain pipeline.

**Figure 3 brainsci-15-00914-f003:**
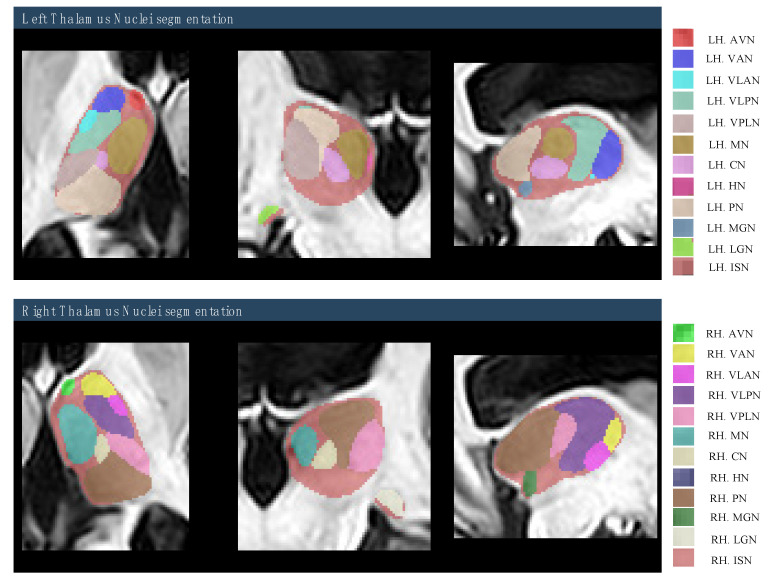
The labeling protocol of thalamic nuclei that are based on Morel’s stereotactic atlas of the human thalamus. The right panel shows the color-coded thalamic nuclei that are overlaid onto the subject’s high-resolution T1-weighted anatomical scan [subject 05].

**Figure 4 brainsci-15-00914-f004:**
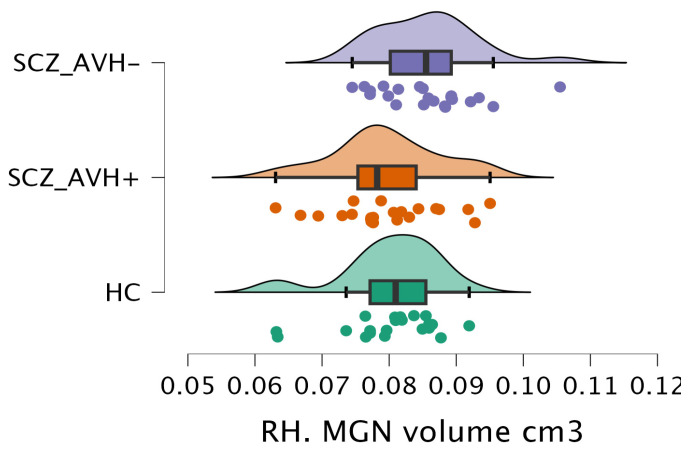
The raincloud charts show the volumetric changes of the RH. MGN between the study group.

**Figure 5 brainsci-15-00914-f005:**
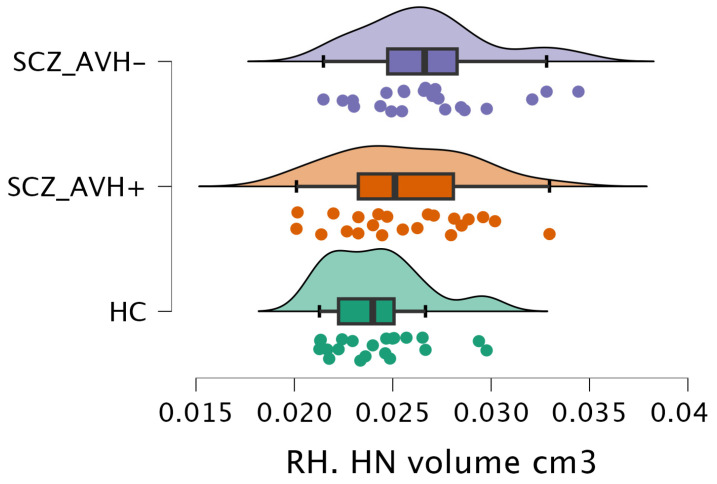
The raincloud charts show the volumetric changes of the RH. HN between the study group.

**Figure 6 brainsci-15-00914-f006:**
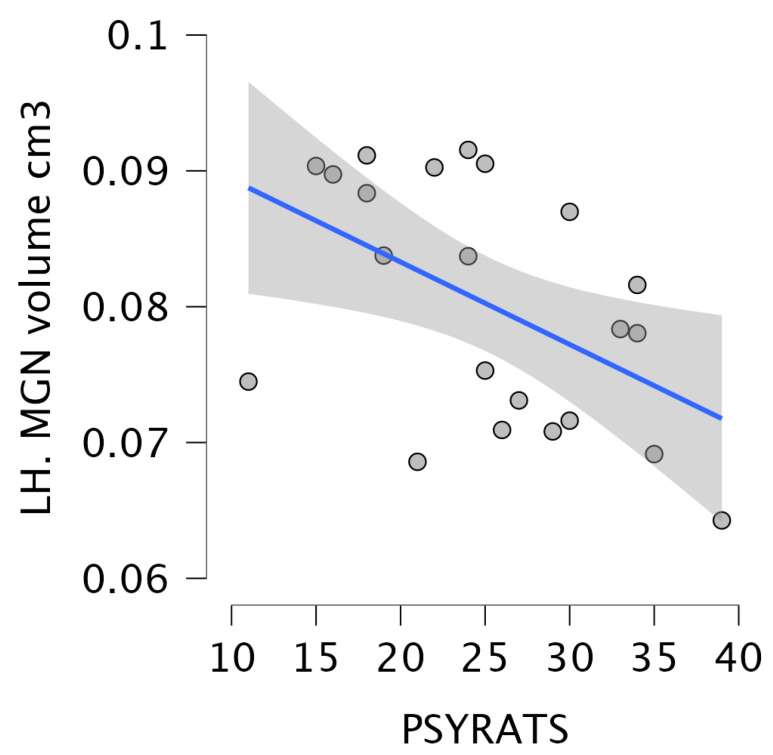
The results of the correlation analysis between the volume of the left MGN and the psychotic ratings. The circles denote the plotted data points, while the blue line indicates the regression line.

**Figure 7 brainsci-15-00914-f007:**
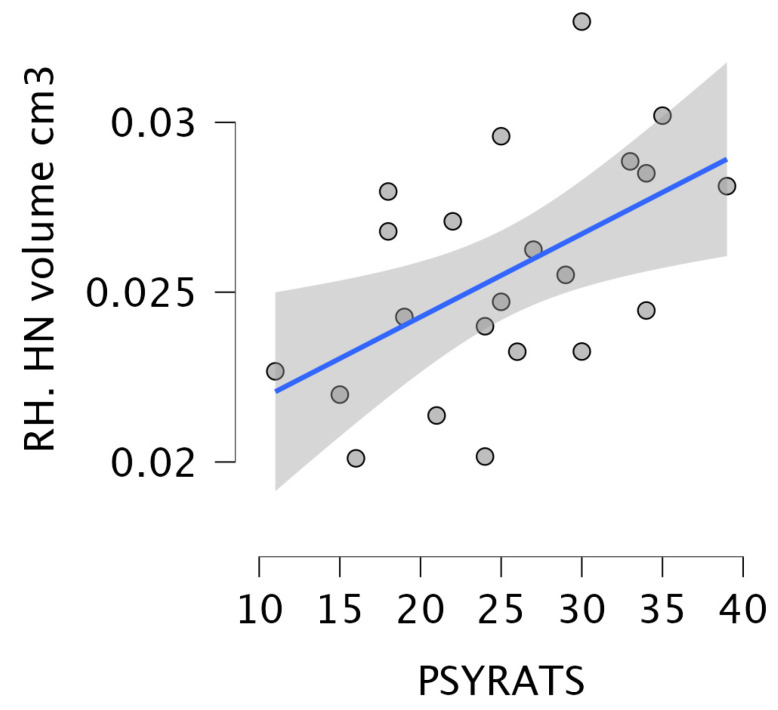
The results of the correlation analysis between the volume of the right HN and the psychotic ratings. The circles denote the plotted data points, while the blue line indicates the regression line.

**Table 1 brainsci-15-00914-t001:** The demographic characteristics of the participants involved in this study.

	HC (*n* = 21)	SZ_AVH− (*n* = 22)	SZ_AVH+ (*n* = 22)	Statistical Value	*p*-Value
Age	38.24 ± 14.43 (31.67–44.8)	45.05 ± 7.54 (41.7–48.39)	39.18 ± 12.5 (33.64–44.73)	F=2.12	0.12
Gender	15M/6F	15M/7F	19M/3F	Χ^2^ = 2.22	0.32
IQ	100.38 ± 10.3 (95.69–105.07)	101.32 ± 8.89 (97.38–105.26)	97.86 ± 9.48 (93.66–102.07)	F = 0.77	0.46
PSYRATS	N/A	N/A	25.23 ± 7.34 (21.97–28.48)	N/A	N/A

Numbers in brackets represent 95% Confidence Interval (CI) limits from the mean of the demographic characteristics.

**Table 2 brainsci-15-00914-t002:** The ANOVA results of the total ICV volumetric analysis across study groups.

	HC (*n* = 21)	SZ_AVH− (*n* = 22)	SZ_AVH+ (*n* = 22)	Statistical Value	*p*-Value
ICV (cm^3^)	1436.12 ± 131.21 (1376.39–1495.85)	1363.51 ± 130.36 (1305.71–1421.31)	1391.7 ± 82.15 (1355.28–1428.12)	F = 2.11	0.13

Numbers in brackets represent 95% Confidence Interval (CI) limits from the mean of the ICV measurements.

**Table 3 brainsci-15-00914-t003:** The ANOVA results of the thalamic nuclei volumetric analysis across study groups.

	Right Hemisphere					Left Hemisphere				
	HC	SCZ_AVH−	SCZ_AVH+	F	*p*	HC	SCZ_AVH−	SCZ_AVH+	F	*p*
Thalamus	6.3 ± 0.3 (5.6–7.0)	6.2 ± 0.4 (5.5–7.2)	6.2 ± 0.3 (5.5–6.8)	1.0	0.36	6.2 ± 0.34 (5.5–6.9)	6.0 ± 0.4 (5.4–7.0)	6.0 ± 0.4 (5.3–6.6)	1.8	0.16
AVN	0.1 ± 0.01 (0.07–0.1)	0.1 ± 0.01 (0.05–0.1)	0.1 ± 0.01 (0.08–0.1)	1.3	0.2	0.1 ± 0.01 (0.08–0.13)	0.1 ± 0.01 (0.07–0.13)	0.1 ± 0.01 (0.05–0.13)	0.4	0.65
VAN	0.28 ± 0.02 (0.2–0.3)	0.27 ± 0.02 (0.2–0.3)	0.28 ± 0.03 (0.2–0.3)	0.2	0.79	0.31 ± 0.02 (0.2–0.3)	0.31 ± 0.03 (0.2–0.3)	0.32 ± 0.03 (0.2–0.3)	0.8	0.42
VLAN	0.1 ± 0.01 (0.1–0.11)	0.1 ± 0.01 (0.1–0.11)	0.1 ± 0.01 (0.1–0.11)	0.2	0.75	0.1 ± 0.01 (0.1–0.11)	0.1 ± 0.01 (0.1–0.11)	0.1 ± 0.01 (0.1–0.11)	0.4	0.65
VLPN	0.89 ± 0.07 (0.85–0.92)	0.85 ± 0.08 (0.81–0.88)	0.84 ± 0.08 (0.8–0.88)	2.1	0.13	0.87 ± 0.05 (0.85–0.9)	0.86 ± 0.07 (0.83–0.89)	0.85 ± 0.08 (0.81–0.88)	0.7	0.47
VPLN	0.35 ± 0.03 (0.34–0.37)	0.34 ± 0.03 (0.33–0.36)	0.35 ± 0.03 (0.34–0.36)	0.4	0.64	0.36 ± 0.04 (0.34–0.38)	0.35 ± 0.04 (0.33–0.37)	0.36 ± 0.03 (0.34–0.37)	0.2	0.80
PN	1.33 ± 0.09 (1.29–1.37)	1.29 ± 0.13 (1.23–1.35)	1.27 ± 0.13 (1.22–1.33)	1.3	0.26	1.43 ± 0.1 (1.38–1.47)	1.37 ± 0.14 (1.31–1.43)	1.37 ± 0.13 (1.31–1.43)	1.5	0.22
LGN	0.09 ± 0.02 (0.08–0.09)	0.08 ± 0.02 (0.07–0.09)	0.08 ± 0.02 (0.07–0.09)	0.6	0.51	0.09 ± 0.02 (0.08–0.1)	0.09 ± 0.02 (0.08–0.09)	0.08 ± 0.02 (0.08–0.09)	1.9	0.15
MGN	0.08 ± 0.01 (0.08–0.08)	0.09 ± 0.01 (0.08–0.09)	0.08 ± 0.01 (0.08–0.08)	4.2	0.01	0.08 ± 0.01 (0.08–0.08)	0.08 ± 0.01 (0.08–0.09)	0.08 ± 0.01 (0.08–0.08)	2.2	0.11
CN	0.13 ± 0.01 (0.13–0.14)	0.13 ± 0.02 (0.12–0.14)	0.13 ± 0.01 (0.13–0.14)	0.3	0.73	0.13 ± 0.01 (0.12–0.13)	0.13 ± 0.01 (0.12–0.13)	0.13 ± 0.01 (0.12–0.14)	0.6	0.53
MN	0.67 ± 0.04 (0.65–0.69)	0.67 ± 0.07 (0.63–0.7)	0.65 ± 0.05 (0.62–0.67)	1.2	0.3	0.66 ± 0.03 (0.65–0.68)	0.66 ± 0.06 (0.64–0.69)	0.65 ± 0.05 (0.63–0.67)	0.3	0.7
HN	0.02 ± 0 (0.02–0.03)	0.03 ± 0 (0.03–0.03)	0.03 ± 0 (0.02–0.03)	3.7	0.03	0.03 ± 0 (0.02–0.03)	0.03 ± 0 (0.03–0.03)	0.03 ± 0 (0.02–0.03)	2.1	0.12

Numbers in brackets represent 95% Confidence Interval (CI) limits from the mean of the thalamus and thalamic nuclei volumes.

**Table 4 brainsci-15-00914-t004:** The post hoc comparison results of the thalamic nuclei volumetric analysis in RH. MGN and RH. HN between study groups.

	Comparisons	t	*p*	95% CI Limits	
				Lower	Upper
RH. MGN	HC vs. SCZ_AVH−	−2.39	0.06	−0.01	0
	HC vs. SCZ_AVH+	0.23	1	−0.01	0.01
	SCZ_AVH− vs. SCZ_AVH+	2.64	0.03	0	0.01
RH. HN	HC vs. SCZ_AVH−	−2.73	0.02	0	0
	HC vs. SCZ_AVH+	−1.43	0.4	0	0
	SCZ_AVH− vs. SCZ_AVH+	1.32	0.5	0	0

**Table 5 brainsci-15-00914-t005:** The ANOVA results of the Asymmetry Index analysis of the thalamic nuclei across study groups.

	Asymmetry Index (AI)				
	HC	SCZ_AVH−	SCZ_AVH+	F	*p*
Thalamus	0.002 ± 0.08 (−0.03–0.03)	−0.05 ± 0.08 (−0.09–−0.01)	−0.03 ± 0.05 (−0.06–−0.01)	3.07	0.054
AVN	−0.03 ± 0.1 (−0.08–0.008)	−0.03 ± 0.1 (−0.08–0.01)	−0.07 ± 0.15 (−0.14–−0.01)	0.94	0.39
VAN	0.1 ± 0.1 (0.06–0.15)	0.07 ± 0.12 (0.01–0.12)	0.11 ± 0.1 (0.06–0.15)	0.87	0.42
VLAN	0.01 ± 0.11 (−0.03–0.06)	0.03 ± 0.08 (0.003–0.07)	0.04 ± 0.09 (0.005–0.08)	0.45	0.63
VLPN	−0.01 ± 0.05 (−0.03–0.01)	0.01 ± 0.04 (−0.005–0.03)	0.01 ± 0.04 (−0.009–0.02)	1.63	0.2
VPLN	0.009 ± 0.05 (−0.01–0.03)	0.02 ± 0.05 (0.0008–0.04)	0.02 ± 0.05 (−0.005–0.04)	0.44	0.64
PN	0.07 ± 0.03 (0.05–0.08)	0.06 ± 0.04 (0.04–0.08)	0.07 ± 0.04 (0.05–0.09)	0.64	0.52
LGN	0.09 ± 0.17 (0.01–0.17)	0.11 ± 0.22 (0.01–0.2)	0.02 ± 0.15 (−0.04–0.09)	1.17	0.31
MGN	−0.01 ± 0.1 (−0.06–0.03)	−0.02 ± 0.07 (−0.05–0.01)	0.04 ± 0.08 (−0.03–0.04)	0.49	0.61
CN	−0.03 ± 0.08 (−0.07–0.001)	0.005 ± 0.09 (−0.03–0.04)	−0.004 ± 0.08 (−0.04–0.03)	1.34	0.27
MN	−0.009 ± 0.03 (−0.02–0.008)	−0.0001 ± 0.04 (−0.02–0.02)	0.01 ± 0.04 (−0.01–0.03)	1.13	0.32
HN	0.03 ± 0.11 (−0.01–0.08)	0.002 ± 0.1 (−0.04–0.04)	0.01 ± 0.11 (−0.03–0.07)	0.51	0.6

Numbers in brackets show 95% Confidence Interval (CI) limits for the mean of the Asymmetry Index (AI) of the thalamus and thalamic nuclei volumes.

**Table 6 brainsci-15-00914-t006:** The results of regression model analysis show the association between thalamic nuclei volumetric measurements (dependent variable) and demographic characteristics (independent variables).

	Regression Model			Age		Gender		IQ		Study Group (SCZ_AVH−)		Study Group (SCZ_AVH+)	
	df	F	*p*	β	*p*	β	*p*	β	*p*	β	*p*	β	*p*
Thalamus	5	4.64	0.001	−0.31	0.01	0.26	0.02	−0.26	0.02	−0.12	0.35	−0.2	0.13
AVN	5	1.24	0.29	−0.12	0.34	0.09	0.5	−0.2	0.13	−0.05	0.72	0.1	0.48
VAN	5	0.75	0.59	−0.02	0.85	0.09	0.51	−0.17	0.18	−0.07	0.65	0.06	0.71
VLAN	5	0.85	0.51	0.15	0.24	0.16	0.22	−0.05	0.69	−0.09	0.56	0.08	0.58
VLPN	5	2.59	0.03	−0.22	0.07	0.29	0.02	−0.09	0.46	−0.11	0.44	−0.2	0.16
VPLN	5	1.69	0.14	−0.3	0.01	0.1	0.42	0.19	0.14	−0.05	0.73	−0	0.99
PN	5	3.97	0.003	−0.36	0.003	0.19	0.1	−0.21	0.08	−0.1	0.48	−0.21	0.11
LGN	5	2.1	0.07	−0.08	0.5	0.33	0.01	−0.03	0.78	−0.18	0.21	−0.15	0.30
MGN	5	1.8	0.12	0.06	0.65	−0.09	0.45	−0.13	0.29	0.34	0.02	−0.02	0.9
CN	5	2.27	0.05	−0.2	0.11	−0.07	0.56	−0.34	0.009	0.07	0.64	0.06	0.67
MN	5	2.35	0.05	−0.18	0.16	0.1	0.43	−0.3	0.01	0.03	0.81	−0.18	0.19
HN	5	2.94	0.01	0.02	0.85	0.21	0.08	0.26	0.03	0.36	0.01	0.27	0.05

## Data Availability

The original contributions presented in this study are included in the article. Further inquiries can be directed to the corresponding author.

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
