# Peer review of "The Involvement of Thalamic Nuclei in Schizophrenia Patients with and Without Auditory Verbal Hallucinations: A Brain Morphometry Study"

_brainsci, 2025, doi:10.3390/brainsci15090914_

Round 1

Reviewer 1 Report

Comments and Suggestions for Authors

Dear authors, thanks for sharing with us reviewers this interesting manuscript. While the overall design is solid, the discussion and methods should be partially reframed. Given the limitations of the study (see my detailed report below), the discussion should be largely rewritten to better account for them.

Introduction

  • The introduction lacks a cohesive narrative or central idea that justifies the study. The motivation behind focusing on thalamic nuclei in relation to AVHs in schizophrenia should be clarified and integrated throughout.

  • Several statements are presented as established facts when they are, in fact, hypotheses. These should be appropriately qualified.

  • The citation “It affects 35 approximately 24 million people, or 1 in 300 worldwide (WHO, 2022)” uses an incorrect reference format and should be corrected.

Methods

  • The absence of medication data is a significant limitation and must be explicitly acknowledged in the limitations section or corrected if available.

  • There is no mention of a power analysis for the chosen sample size; this should be addressed.

  • The cross-sectional design limits causal inference and should be clearly stated in the limitations.

  • The rationale for selecting specific thalamic nuclei appears arbitrary; justification should be improved.

Discussion

  • The claim that MGN volume may serve as a biomarker is overstated, especially considering the cross-sectional design. The discussion should be more cautious.

  • The discussion lacks depth in contextualizing the findings within the broader literature.

  • While age and IQ are considered, other key confounding variables—such as medication status, illness duration, and substance use—are not adequately addressed. This should at the very least be discussed.

Author Response

Author's Reply to the Review Report (Reviewer 1)

Reviewer 1

Dear authors, thanks for sharing with us reviewers this interesting manuscript. While the overall design is solid, the discussion and methods should be partially reframed. Given the limitations of the study (see my detailed report below), the discussion should be largely rewritten to better account for them.

Comment 1: The introduction lacks a cohesive narrative or central idea that justifies the study. The motivation behind focusing on thalamic nuclei in relation to AVHs in schizophrenia should be clarified and integrated throughout.

Response 1: Thank you for pointing this out. I agree with your comment. Therefore, I have provided the reasoning behind focusing on thalamic nuclei regarding AVHs in schizophrenia, as detailed on page 2, lines 87-90. 

Comments 2: Several statements are presented as established facts when they are, in fact, hypotheses. These should be appropriately qualified.

Response 2: Thank you for pointing this out. I agree with your comment. Therefore, I have amended these statements, which can be found on page 1, lines 27-30 and 38, and on page 2, lines 46-47 and 87-88.

Comments 3: The citation “It affects 35 approximately 24 million people, or 1 in 300 worldwide (WHO, 2022)” uses an incorrect reference format and should be corrected.

Response 3: Thank you for pointing this out. I agree with your comment. Therefore, I have corrected this citation found on page 1, lines 35-36.

Comments 4: The absence of medication data is a significant limitation and must be explicitly acknowledged in the limitations section or corrected if available.

Response 4: Thank you for pointing this out. I agree with your comment. Therefore, I have acknowledged in the limitation section that can be found on page 13, lines 381-389.

Comments 5: There is no mention of a power analysis for the chosen sample size; this should be addressed.

Response 5: Thank you for pointing that out. I agree with your comment. Therefore, I have addressed this point in the limitation section, which can be found on page 13, lines 387-389.

Comments 6: The cross-sectional design limits causal inference and should be clearly stated in the limitations.

Response 6: Thank you for pointing that out. I agree with your comment. Therefore, I have addressed this point in the limitations section, which can be found on page 13, lines 387-389.

Comments 7: The rationale for selecting specific thalamic nuclei appears arbitrary; justification should be improved.

Response 7: Thank you for pointing that out. The labeling protocol employed the Morel stereotactic atlas of the human thalamus, which is based on multiarchitectonic parcellation.

Comments 8: The claim that MGN volume may serve as a biomarker is overstated, especially considering the cross-sectional design. The discussion should be more cautious.

Response 8: Thank you for pointing that out. I agree with your comment. Therefore, I have amended this statement that can be found on page 1, lines 27-29, and page 13, lines 394-396.

Comments 9: The discussion lacks depth in contextualizing the findings within the broader literature.

Response 9: Thank you for pointing that out. I agree with your comment. I have revised the discussion for better cohesion, which can now be found on page 12, lines 322-324.

Comments 10: While age and IQ are considered, other key confounding variables—such as medication status, illness duration, and substance use—are not adequately addressed. This should at the very least be discussed.

Response 10: Thank you for pointing that out. I agree with your comment. I have addressed this point, which can now be found on page 13, lines 372-380.

Reviewer 2 Report

Comments and Suggestions for Authors

This study aims to investigate whether volumetric differences in thalamic nuclei (with focus on the MGN and HN, and lateralization) can distinguish schizophrenia patients with auditory verbal hallucinations (AVH) from those without. Using high-resolution MRI data from an open-access dataset and automated segmentation methods, the study compares thalamic subregion volumes across patient and control groups and examines their correlation with psychotic symptom severity. A key strength of the manuscript is its detailed anatomical focus on thalamic subnuclei, inclusion of lateralization analysis, and use of transparent, reproducible methods.  A drawback is the description of the quantities used in the methods is ambiguous in places.  The findings are consistent with previous literature and contribute to the growing understanding of thalamocortical involvement in AVH.

General comments:

Clarity and relevance:

The manuscript is generally clear and written in an accessible manner. There is sufficient detail in the rationale and the methods, with key results communicated effectively. There are occasional and minor lapses in tone. The paper is relevant for the field as it addresses an important question in schizophrenia research in relation to the thalamic nuclei and other neurobiological investigations. It is well-structured, following the standard scientific format, as well as being logical in flow. The manuscript does identify a gap in the literature regarding the role of thalamic nuclei in schizophrenia patients with and without auditory verbal hallucinations. However, the gap could be more explicitly stated in the introduction. Perhaps by directly noting the lack of prior studies comparing lateralization indices.

References:

 17 of the 42 references are recent. This might be acceptable because of the inclusion of the relevant foundational studies. There were no self-citations.

Experimental design:

The research question is clearly stated - assessing whether volumetric changes in thalamic nuclei distinguishes schizophrenia with AVH from those without. The use of structural MRI and volumetric segmentation, using appropriate pre-processing methods, is a suitable method for testing the hypothesis. The study includes three well-defined groups, of moderate size (mentioned as a limitation). All necessary information about the patient screening was included. Clear background and helpful anatomical context is included, with context given on the thalamus and subregions. Necessary statistical analyses were used (correlations, ANOVA, regression, post hoc tests). Overall, the design supports the conclusion.

Reproducibility:

The results seem to be reproducible as the data is extracted from an open source repository (OpenNeuro). The level of detail in the Methods section is adequate. All processes are clearly described: data acquisition, consent, inclusion and exclusion criteria, MRI parameters are specifically mentioned with full detail, and the full processing pipeline (including details of software used and segmentation steps).

Figures and Tables:

The figures and tables in the manuscript are appropriate and generally clear in their presentation. While they are a bit cramped, all information is conveyed. The authors use tables to present group comparisons of volumetric data across thalamic nuclei, and the statistical results are reported with mean values, standard deviations, F-values, and p-values. These tables are well-formatted and help convey group differences and statistical significance. The use of raincloud plots for volumetric comparisons and asymmetry indices is helpful. They effectively show both the distribution and summary of the data. The figures use contrasting colors, so the visualization and differentiability of brain regions is visible. The figures and tables match the results described in the text, and the visual presentation is consistent with the interpretation of findings. The statistical analyses (ANOVA, Bonferroni post hoc tests, correlations, and regression models) are appropriate for the study design, and the use of normalized volumes and correction for intracranial volume is explained clearly. The use of a publicly available dataset (OpenNeuro) is also clearly documented, along with the MRI acquisition parameters.

Conclusions:

The conclusion is consistent with the evidence and the arguments. The conclusion includes areas of further research, as well as potential treatment options. The discussion is well-supported by relevant literature as all findings are largely consistent with the previous research on the thalamus in schizophrenia and auditory hallucinations.

Data availability:

The data availability statement is stated within the article. There is an “Informed Consent Statement" as well as the author’s declaration of no conflicts of interest. The author discloses that they have used OpenAI (line 413).

Literature overlap:

A comparable study, conducted by Pérez-Rando et al. (2022), found several volumetric alterations in thalamic nuclei in schizophrenia patients with AVH, and included PSYRATS scores, however they did not analyze lateralization. https://www.sciencedirect.com/science/article/pii/S2213158222001358

Specific comments:

Abstract, Line 22; Results, Line 244 - Volume units are missing.  Carefully check manuscript throughout as several omissions were found but there may be additional locations.

Introduction, Line 46 - Too casual for scientific tone

Introduction, Lines 72 - 85 - Non-matching order of functional descriptions (flow seems unnatural)

Introduction, Lines 88-90 - Repeats the essence of lines 69-71.  Edit to remove redundancy.    

Methods, Line 137 - Missing citation for volBrain (Manjón JV and Coupé P (2016) volBrain: An Online MRI Brain Volumetry System. Front. Neuroinform. 10:30. doi: 10.3389/fninf.2016.00030.)

Methods, Line 139 - Missing citation for Morel stereotactic atlas. Niemann, K., et al. "The Morel stereotactic atlas of the human thalamus: atlas-to-MR registration of internally consistent canonical model." Neuroimage 12.6 (2000): 601-616.

Figure 1 - Please include the unlabelled MRI image for reference. 

Methods, Line 148 - ANTs is not written out on first use.  Should also indicate the version/release number and any relevant citations.

Methods, Line 157 - Please indicate the basis of the quality control metric, and the exclusion criteria or thresholds for data rejection.

Methods, Line 176, ICV Normalization Methods. I find the calculation of the adjusted volume to be non-intuitive.  Vol_adj is simply Vol_raw*ICV_mean/ICV_raw.  Specify in the definition of beta that it is the slope of the regression line between ROI_raw and ICV_raw.  An improved result would be achieved if beta and ICV_mean were computed only from Healthy controls. 

Methods, Line 193 - Equation would be clearer if the factor of 2 was shown on the leftmost side.  There is ambiguity between “Adjusted volumes” and “Normalized volumes”, please be consistent.  Additionally, the raw volumes can be used for the laterality computation, there is no need to scale first to the normalized space.

Methods, Line 207 - Include the meaning of the test value of 27. 

Methods, Line 230 - If adjusted volumes were used here, state it explicitly. 

Table 2 - The mean ICV for each group should also be reported.   

Results, Line 251 - Remove the phrase, “borderline significant”, it is not meaningful.  Additionally, if the relationship was not significant, it should not be stated that the values “demonstrated” laterality in the following lines. 

Results, Line 306 - Do you mean 12 model parameters?  12 models does not make sense to me, I see a single multiple regression model. 

Figure 4 - It would be more intuitive to see a linear fit to the data adjusted for the multiple parameters.

Discussion, line 324 - Cannot state that lateralization was observed if result was not significant. 

Discussion, line 380 - “indicating that” should be replaced with “and” or “so”.    

Author Response

Author's Reply to the Review Report (Reviewer 2)

Reviewer 2

This study aims to investigate whether volumetric differences in thalamic nuclei (with focus on the MGN and HN, and lateralization) can distinguish schizophrenia patients with auditory verbal hallucinations (AVH) from those without. Using high-resolution MRI data from an open-access dataset and automated segmentation methods, the study compares thalamic subregion volumes across patient and control groups and examines their correlation with psychotic symptom severity. A key strength of the manuscript is its detailed anatomical focus on thalamic subnuclei, inclusion of lateralization analysis, and use of transparent, reproducible methods.  A drawback is the description of the quantities used in the methods is ambiguous in places.  The findings are consistent with previous literature and contribute to the growing understanding of thalamocortical involvement in AVH.

Comments 1: The manuscript does identify a gap in the literature regarding the role of thalamic nuclei in schizophrenia patients with and without auditory verbal hallucinations. However, the gap could be more explicitly stated in the introduction. Perhaps by directly noting the lack of prior studies comparing lateralization indices.

Response 1: Thank you for pointing this out. I agree with your comment. Therefore, I have noted the lack of prior studies comparing lateralization indices, as detailed on page 3, lines 93-95.

Comments 2: A comparable study, conducted by Pérez-Rando et al. (2022), found several volumetric alterations in thalamic nuclei in schizophrenia patients with AVH, and included PSYRATS scores, however they did not analyze lateralization.

Response 2: Thank you for highlighting this. I agree with your comment. As a result, I have included this comparison study in the discussion section, which is on page 12, lines 316-318. 

Comments 3: Abstract, Line 22; Results, Line 244 - Volume units are missing.  Carefully check manuscript throughout as several omissions were found but there may be additional locations.

Response 3: Thank you for pointing this out. I agree with your feedback and have now included the volume units throughout the manuscript.

Comments 4: Introduction, Line 46 - Too casual for scientific tone

Response 4: Thank you for highlighting this matter. I concur with your observation. Consequently, I have amended the statement located on page 2, lines 45-47.

Comments 5: Introduction, Lines 72 - 85 - Non-matching order of functional descriptions (flow seems unnatural)

Response 5: Thank you for pointing this out. I agree with your observation. As a result, I have aligned the order of the functional descriptions found on page 2, lines 72-86.

Comments 6: Introduction, Lines 88-90 - Repeats the essence of lines 69-71.  Edit to remove redundancy.    

Response 6: Thank you for highlighting this. I concur with your observation and have revised the document by editing the content on page 2, lines 69-71 and 88-90.

Comments 7: Methods, Line 137 - Missing citation for volBrain (Manjón JV and Coupé P (2016) volBrain: An Online MRI Brain Volumetry System. Front. Neuroinform. 10:30. doi: 10.3389/fninf.2016.00030.)

Response 7: Thank you for pointing this out. I have included the citation on page 4, line 145.

Comments 8: Methods, Line 139 - Missing citation for Morel stereotactic atlas. Niemann, K., et al. "The Morel stereotactic atlas of the human thalamus: atlas-to-MR registration of internally consistent canonical model." Neuroimage 12.6 (2000): 601-616.

Response 8: Thank you for pointing this out. I have included the citation on page 4, line 148.

Comments 9: Figure 1 - Please include the unlabelled MRI image for reference. 

Response 9: Thank you for pointing this out. I have included the unlabeled MRI image for reference on page 4, line 140.

Comments 10: Methods, Line 148 - ANTs is not written out on first use.  Should also indicate the version/release number and any relevant citations.

Response 10: Thank you for pointing this out. I have spelled out the abbreviations, indicated the version, and included the citation on page 4, line 158.

Comments 11: Methods, Line 157 - Please indicate the basis of the quality control metric, and the exclusion criteria or thresholds for data rejection.

Response 11: Thank you for highlighting this. I have detailed the visual quality control procedure on page 5, lines 180-185. 

Comments 12: Methods, Line 176, ICV Normalization Methods. I find the calculation of the adjusted volume to be non-intuitive.  Vol_adj is simply Vol_raw*ICV_mean/ICV_raw.  Specify in the definition of beta that it is the slope of the regression line between ROI_raw and ICV_raw.  An improved result would be achieved if beta and ICV_mean were computed only from Healthy controls. 

Response 12: Thank you for pointing that out. I have corrected the ICV Normalization method on page 5, lines 186-193. 

Comments 13: Methods, Line 193 - Equation would be clearer if the factor of 2 was shown on the leftmost side.  There is ambiguity between “Adjusted volumes” and “Normalized volumes”, please be consistent.  Additionally, the raw volumes can be used for the laterality computation, there is no need to scale first to the normalized space.

Response 13: Thank you for highlighting that. I have revised the Asymmetry Index (AI) equation on page 6, lines 195-204. 

Comments 14:Methods, Line 207 - Include the meaning of the test value of 27. 

Response 14: Thank you for highlighting that. I have removed this one-sample t—test analysis.

Comments 15:Methods, Line 230 - If adjusted volumes were used here, state it explicitly. 

Response 15: Thank you for pointing that out. I mentioned this on page 7, lines 238-239.

Comments 16: Table 2 - The mean ICV for each group should also be reported.   

Response 16: Thank you for pointing that out. I reported the mean ICV in Table 2, on page 7, line 235.

Comments 17: Results, Line 251 - Remove the phrase, “borderline significant”, it is not meaningful.  Additionally, if the relationship was not significant, it should not be stated that the values “demonstrated” laterality in the following lines. 

Response 17: I appreciate your feedback. The phrase on page 9, lines 264-269, has been removed.

Comments 18: Results, Line 306 - Do you mean 12 model parameters?  12 models does not make sense to me, I see a single multiple regression model. 

Response 18: Thank you for your feedback. I have corrected the description of the regression model on page 11, lines 295-301.

Comments 19: Figure 4 - It would be more intuitive to see a linear fit to the data adjusted for the multiple parameters.

Response 19: Thank you for your feedback. I have made the changes in figures 6 and 7 on pages 10 and 11.

Comments 20: Discussion, line 324 - Cannot state that lateralization was observed if result was not significant. 

Response 20: Thank you for your feedback. The statement has been removed from page 12.

Comments 21: Discussion, line 380 - “indicating that” should be replaced with “and” or “so”.    

Response 21: Thank you for your feedback. The statement has been updated on page 13, line 379.

Reviewer 3 Report

Comments and Suggestions for Authors

Dear Editor-In-Chief

Brain Science, MDPI,

Subject: Review of the article brainsci-3811371

Entitled “The Involvement of Thalamic Nuclei in Schizophrenia Patients with and without Auditory Verbal Hallucinations: A Brain Morphometry Study”

This study aims to evaluate the volume of thalamic nuclei in two groups of individuals with schizophrenia, without and with auditory verbal hallucination. The study is interesting and relevant to the journal scope. The work can be further enhanced. Enclosed below are some general and specific comments for the authors to consider.

General comments

  1. how to confidently relate age changes in thalamus volume across to schizophrenia?
  2. What would be the ultimate application of such studies? Does it contribute to the diagnosis or the therapy management?

Specific comments:

  1. line 23 & 24: if t refers to the Person coefficient test, then it is usually referred to by r. otherwise, clarify.
  2. line 24: spell out MGN
  3. line 136: is the MRI brain volumetry software FDA approved?
  4. line 138: since the thalamus pipeline was used to segment the thalamic nuclei, this is a semi-automated approach not fully automated.
  5. Pre-processing: a diagram to illustrate the pre-processing steps including fusion, and cropping is needed.
  6. line 162: how did you obtain the quality control score? Tools, professionals and methods needs to be addressed for reproducibility.
  7. ICV normalization: raw data means non-reconstructed image, hence how was the ICV raw determined in a raw data and cm3? Number of voxels would have been more visible.
  8. lines 207 & 208: since 27 subjects’ data from SCZ_AVH+ used in previous studies were enrolled, this needs to be clearly stated in section 2.1.
  9. line 221: what is (F(2,62) referring to? What does this symbol refer to η² likewise w?
  10. table 2 needs better formatting, the present format reads confusing.
  11. table 3, remove the negative sign next to zero and keep consistent use of the dash symbol “-“. The same applies to all tables whenever applies.
  12. For consistency, the p precision should be consistent across the manuscript, either 2 or 3 decimals.
  13. line 248 & 282: groups instead of group
  14. citation format: He et [2021] instead of He et al (2021). Amend this across the manuscript.

Author Response

Author's Reply to the Review Report (Reviewer 3)

Reviewer 3

Entitled “The Involvement of Thalamic Nuclei in Schizophrenia Patients with and without Auditory Verbal Hallucinations: A Brain Morphometry Study”

This study aims to evaluate the volume of thalamic nuclei in two groups of individuals with schizophrenia, without and with auditory verbal hallucination. The study is interesting and relevant to the journal scope. The work can be further enhanced. Enclosed below are some general and specific comments for the authors to consider.

Comments 1: how to confidently relate age changes in thalamus volume across to schizophrenia?

Response 1: Thank you for pointing that out. The multiple regression analysis showed no significant interaction between age and study groups regarding changes in volumetric measurements. This indicates that the effects of age and study group on volumetric measurements can be viewed separately.

Comments 2: What would be the ultimate application of such studies? Does it contribute to the diagnosis or the therapy management?

Response 2: Thank you for pointing that out. This study aims to analyze the brain structure of the thalamic nuclei in schizophrenia patients experiencing auditory verbal hallucinations (AVHs).

Comments 3: line 23 & 24: if t refers to the Person coefficient test, then it is usually referred to by r. otherwise, clarify.

Response 3: Thank you for highlighting that. I have amended this on page 1, lines 22 and 23. 

Comments 4: line 24: spell out MGN

Response 4: Thank you for highlighting that. I have spelled this out on page 1, lines 20 and 25. 

Comments 5: line 136: is the MRI brain volumetry software FDA approved?

Response 5: Thank you for highlighting that. This platform is free to use for non-commercial and non-medical purposes, such as research only.

Comments 6: line 138: since the thalamus pipeline was used to segment the thalamic nuclei, this is a semi-automated approach not fully automated.

Response 6: Thank you for highlighting that. The volBrain platform functions completely automatically and can deliver brain analysis without human input.

Comments 7: Pre-processing: a diagram to illustrate the pre-processing steps including fusion, and cropping is needed.

Response 7: Thank you for pointing that out. I have included the pre-processing diagram in figure 2, on page 5, line 165. 

Comments 8: line 162: how did you obtain the quality control score? Tools, professionals and methods needs to be addressed for reproducibility.

Response 8: Thank you for pointing that out. I explained the quality control procedure on page 6, lines 181-186. 

Comments 9: ICV normalization: raw data means non-reconstructed image, hence how was the ICV raw determined in a raw data and cm3? Number of voxels would have been more visible.

Response 9: Thank you for highlighting that. Raw volumes denote the unadjusted measurements of brain regions, prior to applying any normalization techniques.

Comments 10: lines 207 & 208: since 27 subjects’ data from SCZ_AVH+ used in previous studies were enrolled, this needs to be clearly stated in section 2.1.

Response 10: Thank you for highlighting that. I have removed this one-sample t—test analysis.

Comments 11: line 221: what is (F(2,62) referring to? What does this symbol refer to η² likewise w?

Response 11: Thank you for pointing that out. (F(2,62)) indicates the statistical values from the ANOVA test, representing F(df1, df2) with numerator degrees of freedom (df1) and denominator degrees of freedom (df2). η² and w are effect size metrics for numerical and categorical variables, respectively.

Comments 12: table 2 needs better formatting, the present format reads confusing.

Response 12: Thank you for your feedback. I improved the formatting of all tables across the manuscript to enhance clarity and readability.

Comments 13: table 3, remove the negative sign next to zero and keep consistent use of the dash symbol “-“. The same applies to all tables whenever applies.

Response 13: Thank you for your feedback. I removed the negative sign next to zero and ensured consistent use of the dash symbol “-" across all tables where applicable.

Comments 14: For consistency, the p precision should be consistent across the manuscript, either 2 or 3 decimals.

Response 14: Thank you for your feedback. I have revised the format of the P values throughout the manuscript. 

Comments 15: line 248 & 282: groups instead of group

Response 15: Thank you for your feedback. I have corrected the typo.

Round 2

Reviewer 1 Report

Comments and Suggestions for Authors

Thanks to the authors. I have no further objections towards pubblication.